# Deciphering the Mechanism of Tolerance to Apple Replant Disease Using a Genetic Mapping Approach in a Malling 9 × *M. × robusta* 5 Population Identifies SNP Markers Linked to Candidate Genes

**DOI:** 10.3390/ijms24076307

**Published:** 2023-03-27

**Authors:** Stefanie Reim, Ofere Francis Emeriewen, Andreas Peil, Henryk Flachowsky

**Affiliations:** Institute for Breeding Research on Fruit Crops, Julius Kühn Institute (JKI)—Federal Research Centre for Cultivated Plants, Pillnitzer Platz 3a, 01326 Dresden, Germany

**Keywords:** ARD, soil-borne disease, genetic mapping, molecular marker, plant immunity

## Abstract

Apple replant disease (ARD) is a worldwide economic risk in apple production. Although several studies have shown that the wild apple accession *Malus × robusta* 5 (Mr5) is ARD-tolerant, the genetics of this tolerance have not yet been elucidated. A genetic mapping approach with a biparental population derived from contrasting parents involving molecular markers provides a means for marker-assisted selection of genetically complex traits and for determining candidate genes. In this study, we crossed the ARD-tolerant wild apple accession Mr5 and the ARD-susceptible rootstock ‘M9’ and analyzed the resultant progeny for ARD tolerance. Hence, a high-density genetic map using a tunable genotyping-by-sequencing (tGBS) approach was established. A total of 4804 SNPs together with 77 SSR markers were included in the parental maps comprising 17 linkage groups. The phenotypic responses to ARD were evaluated for 106 offspring and classified by an ARD-susceptibility index (ASI). A Kruskal–Wallis test identified SNP markers and one SSR marker on linkage groups (LG) 6 and 2 that correlated with ARD tolerance. We found nine candidate genes linked with these markers, which may be associated with plant response to ARD. These candidate genes provide some insight into the defense mechanisms against ARD and should be studied in more detail.

## 1. Introduction

Apple replant disease (ARD) is a soil-borne disease complex that occurs after replanting apple trees on a site previously planted with apples. Characteristic symptoms of diseased plants include stunted shoot growth and root damage [1,2,3]. This results in reduced and delayed fruit yields and decreased fruit and tree quality [4,5], making this disease an economic risk for fruit growers and tree nurseries worldwide.

Previous cultivation of apples in the same location is thought to cause changes in the soil (micro)biome that affect the ability of the soil to support the subsequent apple crop [3]. Based on recent studies, there is strong evidence that several microorganisms such as *Nectriaceae* [6,7,8], *Pythium* [5], *Rhizoctonia* [5], *Streptomyces* [9], *Actinomycetes* [7,10], and *Fusarium* [11] are involved in ARD as pathogens and/or co-pathogens. The severity of ARD often varies significantly at the regional level but also at a subplot level, which is likely due to differences in soil type and other abiotic and biotic factors [12].

Measures to overcome ARD are often not effective enough, very expensive, or involve strict restrictions to avoid negative environmental effects [3]. The use of ARD-tolerant rootstocks would be a cost-effective and environmentally friendly measure for fruit growers to counteract ARD. Only a few rootstocks with different levels of tolerance have been reported so far [13,14]. However, until now, the widely used rootstock ‘M9’ has not been replaced by such rootstocks, which could be a result of propagation problems, their agronomic performance, compatibility with cultivars, susceptibility to other diseases, or their availability. The breeding of new ARD-tolerant rootstocks requires time and resources. The development of genetic markers to identify ARD-tolerant genotypes can greatly increase breeding efficiency by early selection of progeny with ARD tolerance.

To date, little is known about the genetic mechanisms behind ARD tolerance. Several studies demonstrated expression of biotic stress response genes in plants grown in ARD soils [15,16,17,18,19]. In particular, phytoalexins of the biphenyl biosynthetic pathway have been shown to be activated in response to ARD and have antifungal and antibacterial activity in plants [20]. Phenolic compounds have also been shown to accumulate in response to ARD-infected soil and may act as scavengers of reactive oxygen species (ROS) [21]. Numerous flavonoid metabolism genes as well as genes involved in auxin, ethylene, jasmonate, and cytokinin biosynthesis and signal transduction also appear to be involved in a response against ARD [15,22,23,24,25,26,27]. A comparative transcriptome analysis of Mr5 and ‘M9’ cultivated in ARD soil and disinfected ARD soil identified numerous candidate genes involved in a response to ARD [28]. Several genes identified in this study were involved in cell wall stabilization, were associated with the detoxification processes, were connected with the metabolism of volatile and nonvolatile compounds within the mevalonate (MVA) pathway, or were pathogen-related genes [28]. The latter study suggests that the ARD tolerance in Mr5 possibly requires the activation of numerous defense mechanisms.

Genetic mapping is critical for identifying markers and genes linked to desirable traits, including tolerance to plant pathogens. This approach can be used to localize the genomic position of a certain gene and refine the number of possible candidate genes. SNPs are suitable markers to establish high-density genetic maps because they are the most abundant form of DNA variation and could reach higher density than any other marker types. Genotyping-by-sequencing (GBS), which is a cost-effective method that allows for reliable and robust identification of de novo SNPs, is widely applied in plant species, including members of the Rosaceae family such as apple and its wild relatives [29,30].

The aim of our study was the identification of molecular markers associated with ARD tolerance using progenies of the crossing population ‘M9’ × Mr5. The parent ‘M9’ is the most commonly used rootstock worldwide and known to be susceptible to ARD, whereas the wild apple accession Mr5 is known to be ARD-tolerant [31]. Two high-density parental genetic maps of ‘M9’ and Mr5 were established using de novo SNPs generated by GBS and SSR markers. Using phenotypic data of the progeny following ARD evaluation and the genotypic data for marker–phenotype analyses, thirteen significant SNP markers and one significant SSR marker were detected on chromosomes 6 and 2, which could be linked to putative candidate genes.

## 2. Results

### 2.1. tGBS SNP Identification

From 140 sequenced samples, 397,899,332 quality trimmed reads with an average length of 142 bp were provided after Illumina HiSeq X sequencing by Data2Bio. SNP filtering resulted in a total SNP set containing 169,470 SNPs. From 140 seedlings, 2 progenies were removed due to a high missing rate, so that 138 progenies were included in subsequent analysis. To improve the utility of the selected SNPs, a further cut-off of MCR50 (as described in the methods) was applied, resulting in a subset of 102,662 SNPs. In a further stringent manual quality check, all SNPs with a missing error rate >10%, with bias segregation, and with heterozygous locus in both genotypes (hk × hk) were deleted, which led to 5013 SNPs.

### 2.2. High-Density Linkage Map Construction

Of the 111 tested SSR markers, 89 distributed among the 17 chromosomes of apple were polymorphic and were used together with the 5013 SNPs for the construction of 2 parental maps. Table 1 summarizes the statistics of the established maps. In total, 4804 of the 5103 SNPs and 77 of the 89 SSRs were integrated in the two parental maps (Appendix A).

In the maternal map of ‘M9’, 1879 SNP and 55 SSR markers were mapped over 17 linkage groups (LG). The minimum number of markers per LG was 50 (LG6) and the maximum number was 189 (LG9). The overall genetic length of the ‘M9’ map was 1253.9 cM. In the parental map of Mr5, 2925 SNP markers and 64 of the 89 SSR markers were integrated with a minimum of 83 markers for LG5 and a maximum of 322 markers for LG13. The overall genetic length of the Mr5 map was 1097.7 cM.

### 2.3. Phenotypic Analysis

Shoot length increase was measured from replicates of 106 progeny of the ‘M9’ × Mr5 population, and ASI values were calculated based on these measurements. Negative ASI values indicated progeny with lower susceptibility to ARD, whereas an ASI value of zero or higher indicated higher susceptibility to ARD [31]. ASI values differed significantly, ranging from −11.9 for MAL300026 to 9.2 for MAL300183 (Figure 1; Appendix A). Forty offspring (38%) showed higher shoot length increase (ASI < 0.0) when grown in ARD soil compared with γ-ARD soil, indicating lower susceptibility to ARD under our test conditions. The ARD-tolerant parent Mr5 showed an ASI value of −4.4, confirming the ARD tolerance of this genotype. For the ARD-susceptible parent ‘M9’, the ASI was 2.0, confirming the susceptibility of this genotype. In nine progenies, differences in shoot length between the two soil types were less remarkable with an ASI = 0 to ASI = −1.0. High differences in shoot growth between ARD soil and γ-ARD soil with an ASI ≥ −5.0 were observed in nine progenies. In contrast, in 66 progenies, the ASI was zero or higher than zero because shoot growth on ARD soil was equal to or lower than on irradiated soil, indicating susceptibility to ARD. For 16 progenies, differences in shoot length growth between the two soil types ranged from ASI = 0 to ASI = 1.0. A remarkable higher shoot growth in γ-ARD soil (ASI > 5.0) was observed in 15 offspring. However, the difference in shoot growth between ARD soil and γ-ARD soil was significant in only 26 progenies.

### 2.4. Marker–Phenotype Association Analyses

The established maps for ‘M9’ and Mr5 were employed for genotype–phenotype association by Kruskal–Wallis analysis using the phenotypic data of the 106 progenies, which were based on the ASI value. This analysis, which is a non-parametric equivalent of the one-way ANOVA, quantitatively ranking all individuals and classifying them according to their marker genotype [32], revealed 12 and 2 significant markers on LGs 6 and 2, respectively (Table 2). Of these markers, only CH02c02a maps on the ‘M9’ genetic map, but the allele conferring tolerance is inherited from Mr5. The twelve SNP markers on LG6 of Mr5 are located closely together within the genomic region, spanning a physical region of 794 Kbp between region 26,304,001 and 27,098,135 on chromosome 6. However, interval mapping did not lead to the detection of any QTL on LG6. A typical QTL peak was observed on LG2 with a highest LOD of 2.2, but not exceeding the chromosome threshold of an LOD = 2.7 for a significance level of *p <* 0.05. The sequences of significant SNP markers were aligned to the *Malus domestica* GDDH13 v.1.1 reference genome [33] using Galaxy (https://usegalaxy.org/ accessed on 1 March 2023) in order to identify the genomic position on the *Malus* chromosome. Table 2 shows the markers with the highest significance after the Kruskal–Wallis test and their respective position on the chromosome.

### 2.5. Identification of Potential Candidate Genes in the Malus domestica Genome

Six SNP markers were observed to be located within genes of interest on LG2 and LG6 (Table 3). On LG2, marker SNP_52341 is located within the *calmodulin-binding receptor-like cytoplasmic kinase* 3 gene (*CRK3*; MD02G1022200). On LG6, five SNPs are located within genes that are possibly associated with a response to ARD (Table 3): SNP_56130 is located within the *B3 domain-containing protein Os03g0120900-like* gene (*NGA1*; MD06G1124400), which belongs to the AP2/B3-like transcriptional factor family of proteins. SNP_32246 is located within the *E3 ubiquitin-protein ligase MARCH3* gene (*MARCH3*; MD06G1124900, a membrane-associated RING-CH-type finger protein of E3 ubiquitin ligases. SNP_56970 is located in the apple gene MD06G1125000, which encodes the *dolichol-phosphate mannosyltransferase* gene (*DPM1*). SNP_54395 was found in the apple gene MD06G1125400 that encodes the *walls are thin1*-related protein (*WAT1*). SNP_49388 is located in MD06G1127800, which encodes the *trihelix transcription factor* (*GTL1*). SSR marker Ch02c02a is closely located to a candidate gene on LG2, which encodes a transcription repressor that belongs to the OVATE family protein 8-line (*OFP8*; MD02G1046400). Three more candidate genes are closely located to highly significant SNP markers on LG6 (Table 3): SNP_56298 is closely located to MD06G1103200, an apple gene that encodes an uncharacterized protein. SNP_58943 and SNP_58944 are closely located to the apple gene MD06G1107200, which encodes the *cysteine-rich RLK* (*RECEPTOR-like protein kinase*) *8* gene (*CRK8*). SNP_36646, SNP_36649, and SNP_31883 are closely located to MD06G1123000, which encodes the *E3 ubiquitin-protein ligase RHA2B-like* gene (*BRH1*). SNP_57966 was found to be in close proximity of the apple gene MD06G1125400 that encodes the *walls are thin1*-related protein (*WAT1*).

## 3. Discussion

We employed a genetic mapping approach to identify chromosome regions/genes underlying the tolerance to ARD as this knowledge, including that of the molecular response of apple to ARD, is still scarce. Genetic mapping analysis allows the linkage of a certain trait to specific regions of chromosomes by associating phenotypic and genotypic data. By narrowing down a genetic region, it is possible to search specifically for candidate genes in the genome that could play important roles in tolerance and susceptibility to ARD. Therefore, we performed this study in an attempt to elucidate the genetic basis of this complex trait.

### 3.1. Construction of a High-Density Genetic Map

In the present study, a high-density genetic map of the ARD-susceptible rootstock ‘M9’ and the ARD-tolerant wild apple species *M. × robusta* 5 were constructed using genotyping-by-sequencing SNP technology. This technology enabled the generation of de novo genome-wide SNP data specific to the ‘M9’ and Mr5 genotypes. In total, 4881 markers were successfully mapped. The number of markers mapped in Mr5 (2925 SNPs and 64 SSRs) and the mean interval between adjacent markers (0.37 cM) was remarkably higher than in ‘M9’ (1879 SNPs and 55 SSRs; 0.65 cM). For ‘M9’, the largest gap between two markers (SNP_2125 and SNP_2271) was 26.4 cM on LG7, whereas for Mr5, the largest gap between two markers (NZmsEB142980 and SNP_58346) was 17.3 cM on LG4. The number of markers in the Mr5 map was higher compared to another study in apple with the wild apple species *M. fusca*, which reported 2424 SNPs [29]. Gardiner et al. [30] reported 2436 SNPs for *M. ×domestica*. The main reason for the slightly higher number of genetic markers in Mr5 in this study could be the highly heterozygous genetic background of the population and the fact that the GBS approach did not include a reference genome alignment, which ultimately discards unaligned reads.

### 3.2. Marker–Phenotype Association Analysis

No QTLs were found to exceed the chromosome-wide or genome-wide thresholds based on interval mapping for ARD tolerance. However, SNP markers on LGs 6 and 2 of Mr5 were found to correlate significantly with ARD tolerance. More offspring (62%) were negatively affected by ARD soil and showed an ASI value greater than zero. In contrast, 38% of offspring were not negatively affected by ARD and had better plant growth on ARD soil than on the disinfected soil. Moreover, numerous offspring showed ASI values exceeding or falling below the values of ‘M9’ and Mr5, respectively, which were more pronounced in a tolerant or susceptible response to ARD soil than the parents. Phenotypic data could be influenced by the plant quality after in vitro propagation causing sometimes remarkable variation in plant growth between individual replicates of a genotype. Variations in plant growth may also be explained by the complexity of ARD itself, which is influenced by soil type [34], climate, and other factors, which makes it difficult to classify or evaluate plant responses to ARD [3,31], and possibly by many genetic factors underlying the mechanism of tolerance. However, all bio-tests were performed under defined climatic conditions in the greenhouse, the same soil type was used in every experiment, and all plants were treated equally. Although certainly not all influences could be completely eliminated, notable differences were found in the response of individual progeny to the ARD soil. It is possible that numerous QTL are interacting to deliver tolerance to ARD, with each having little effect.

In a genome-wide transcriptome analysis for the identification of candidate genes associated with tolerance to ARD, several differentially expressed genes were obtained, which were associated with plant stress response, e.g., cell wall stabilization and detoxification or metabolism of volatile and nonvolatile compounds [28]. These findings suggest that the ARD tolerance in Mr5 is based on different pathways. Nevertheless, there was a clear distinct putative effect on LG6 even though the LOD score missed the significant threshold of *p* < 0.05 and putative candidate genes were found within and close to the genomic positions identified by mapping.

### 3.3. Candidate Genes

Six SNP markers were observed to be located within candidate gene sequences on LG2 and LG6 and an additional seven SNPs and one SSR marker were in close proximity to potential candidate genes on LG2 and mainly on LG6. Interestingly, most candidate genes encode for transcription factors (*GTL1*, *NGA1*), transcription repressors (*OFP8*), function as receptors (*CRK8*, *CRCK3*), or belong to the ubiquitin system, which is involved in various cellular processes (*MARCH3*, *BRH1*). The candidate gene *WAT1* functions as a plant drug/metabolite exporter, *DMP1* serves as a donor enzyme, and the *Malus* gene MD06G1103200 was assigned to an uncharacterized protein. Moreover, all candidate genes identified in this study are assumed to be involved in a pathogen response (Figure 2).

#### 3.3.1. Candidate Genes Involved in Plant Innate Immunity

Several SNP markers were within or in close proximity to candidate genes that are involved in plant innate immunity. Plant innate immunity is initiated by pattern recognition receptors (PRRs) at the cell surface that sense conserved molecular signatures of many pathogens, termed microbe-, pathogen-, or damage-associated molecular patterns (MAMPs, PAMPs, DAMPs) [35].

The candidate gene *dolichol-phosphate mannosyltransferase* (*DPM1,* SNP_56970) belongs to an upstream gene that subsequently mediates immune responses via members of the pattern recognition receptors (PRRs). *DPM1* is localized in the endoplasmic reticulum (ER) and is responsible for the synthesis of dolichol P-mannose (Dol-P-Man), which is necessary for the biosynthesis of glycosylphosphatidylinositol-anchored proteins (GPI-APs) [36]. GPI-APs can in turn trigger immune responses due to components of PRRs such as receptor-like kinases (RLKs) or receptor-like proteins (RLPs). It has been found that certain GPI-APs are associated with partner receptor-like kinases (RLKs) [37,38]. In our study, two SNPs (SNP_58943 and SNP_58944) were found to be closely located to the *cysteine-rich RLK (RECEPTOR-like protein kinase) 8* gene (*CRK8*). The exact function of *CRK8* is not known, but in general, CRKs are conserved upstream signaling molecules that play a critical role in the first step of the multistep immune system to protect the plant from pathogens [39]. Another SNP (SNP_32246) was found in the RING-CH ubiquitin ligase gene (*MARCH3*), which is also thought to be involved in plant innate immunity. In mammalian cells, MARCH proteins have been described as regulators of immune responses [40,41], whereas in plants the exact function of MARCH ubiquitin ligases is still unknown. However, plant E3 ubiquitin ligases of the RING-type have generally been shown to play a positive or negative role in regulating various steps of plant immunity [42]. A study in *Arabidopsis* showed that RING-type E3 ubiquitin ligases are induced PAMPs upon pathogen attack, and their activation increased the resistance to the pathogenic soil bacterium *Pseudomonas syringae* [42]. PAMP in turn triggers downstream components for defense activation such as mitogen-activated protein kinase (MAPK) cascades.

Interestingly, in this study, two significant SNP markers were found within genes that play a critical role within the MAPK cascade. SNP_52341 on LG2 was found to be located within *calmodulin-binding receptor-like cytoplasmic kinase 3* (*CRCK3*; MD02G1022200), which is an important signal substrate protein of one of the MAP kinases within the MAPK cascade, namely *MKP4. CRCK3* indirectly triggers an immune response due to Ca^2+^ influx during MAP signaling [43,44]. The second SNP marker (SNP_49388) was located within the *trihelix transcription factor* GT2-like 1 (*GTL1*) on LG6. *GTL1* is a Ca^2+/^calmodulin-binding transcription factor within the MAP kinase cascade [45,46] that positively regulates defense genes and bacterial-triggered immunity [45]. Simultaneously, *GTL1* inhibits factors responsible for plant growth and development, causing plants to exhibit a dwarf phenotype [47]. Interaction studies revealed that genes coordinated by *GTL1* are mainly involved in transport and response but also in salicylic acid (SA) metabolism [45]. These findings lead to us to speculate that these genes involved in plant innate immunity might play a role in an increased tolerance of Mr5 to ARD.

#### 3.3.2. Candidate Genes Involved in Hormone Synthesis

In addition to the initiation of plant innate immunity by PRR, modulation of plant hormone networks plays a central role in regulating immune responses to microbial pathogens, herbivorous insects, and beneficial microbes. The salicylic acid (SA), jasmonic acid (JA), and ethylene (ET) hormone signaling systems form the backbone of the plant immune signaling system, whereas abscisic acid (ABA), auxin, cytokinin, gibberrelin (GA), and brassinosteroid (BR) are involved in modulating plant immune responses by regulating host defense responses triggered by the SA–JA–ET signaling systems [48].

#### 3.3.3. Salicylic Acid (SA) Metabolism

One significant SNP marker was located in a candidate gene which is possibly involved in SA metabolism. SNP_54395 was found within the gene encoding the *walls are thin1*-related protein (*WAT1*). A study in *Arabidopsis* showed that a decrease in *WAT1* expression resulted in dwarfism and a decrease in the thickness of secondary cell walls, as little to no secondary cell walls were formed in fibers [49]. In general, alteration in the structure and/or composition of the plant cell wall, including the secondary cell wall, can lead to changes in disease susceptibility to foliar and soil-borne pathogens [50,51,52]. A transcriptome analysis between ‘M9’ and Mr5 also observed several differentially expressed genes associated with cell wall biosynthesis and hypothesized that ARD-induced cell wall stabilization in Mr5 leads to higher ARD tolerance [28]. However, a later study of *WAT1* expression in *Arabidopsis* provided evidence that cell wall-related mechanisms may not be the direct cause of pathogen resistance, but rather a mechanism involving the detour of tryptophan-derived metabolites into salicylic acid [53]. Therefore, the inactivation of *WAT1*-conferred broad-spectrum resistance to soil-borne pathogens such as *Ralstonia solanacearum, Xanthomonas campestris* pv. *Campestris*, *Verticillium dahlia*, and *Verticillium albo-atrum* [53]. These results suggest that *WAT1* may be involved in tolerance to ARD via SA metabolism in Mr5.

#### 3.3.4. Abscisic Acid (ABA) Biosynthesis

SNP_56130 was located within the *B3 domain-containing protein Os03g0120900-like* gene (*NGA1*/*NGTHA1*, MD06G1124400). *NGA1* belongs to the B3-like transcription factors and plays an important role in regulating cell proliferation and hormone synthesis and transport [54,55]. Studies in *Arabidopsis* and *Medicago truncatula* showed a reduced growth of lateral organs and roots when *NGA1* was overexpressed [56,57]. At the same time, it is assumed that *NGA1* functions as a regulator of stress resistance by activating ABA biosynthesis [57]. ABA is known to confer stress tolerance; however, under prolonged stress and thus high ABA concentrations, plant growth is impaired [58]. These results suggest that the transcription factor *NGA1* may be involved in plant response to ARD, as overexpression of *NGA1* leads to reduced root growth, which was also observed as a phenotypic response of plants to ARD [3].

#### 3.3.5. Brassinosteroid Synthesis

Interestingly, three SNP markers on LG6 and the SSR marker Ch02c02a on LG2 were located near two genes belonging to the brassinosteroid (BR) signaling pathway. SNP_36646, SNP_36649, and SNP_31883 are in close proximity to MD06G1123000 on LG6, which encodes the brassinosteroid-responsive RING-H2 gene (*BRH1*). The SSR marker Ch02c02a was in close proximity to a candidate gene on LG2 encoding a transcription factor of the OVATE family protein 8 lineage (*OFP8*, MD02G1046400), which is also involved in brassinosteroid signaling [59].

In general, brassinosteroids (BRs) are phytohormones that have a variety of effects on plant growth and development, such as short petioles and dwarfism [60,61]. At the same time, BRs play an important role in the trade-offs between growth and defense against pathogens [62]. Here, BR-mediated pathogen defense is induced either by a cell surface-localized receptor kinase (*BAK1*) or *BAK1* independently by the pathogen elicitor chitin. The latter was shown for *BRH1* activation in a study in *Arabidopsis* [63]. One of the major end products of the BR biosynthetic pathway is 24-epibrassinolide (24-epiBL), which has been shown to activate pathogenesis-related genes and protect plants from biotic stress damage, e.g., by *Fusarium* [62,64]. In addition, a study on *Eucalyptus grandis* showed that treatment with 24-epiBL had a significant positive effect on mycorrhizal colonization and the abundance of arbuscular mycorrhiza (AM) fungi in *E. grandis* roots [65]. Whereas 24-epiBL down-regulates *BRH1* expression [66], treatment with 24-epiBL induces *OFP8* expression and results in increased *OFP8* protein accumulation, as shown in a study in rice [59]. The study showed that *OFP8* is a substrate of *glycogen synthase kinase* 2 (*GSK2*, an *Arabidopsis* ortholog of *brassinosteroid insensitive* 2, *BIN2*), a negative regulator that can block BR signaling [59]. The presence of significant SNP and SSR markers in *BRH1* and *OFP8* in Mr5 leads to the suggestion that both candidate genes of the BR signaling pathway may be involved in the ARD tolerance of Mr5. However, their exact functions within the BR signaling pathway are not fully elucidated, and further studies are needed to uncover the possible functions of *BRH1* and *OFP8* in ARD tolerance. Based on these observations, we speculate that both candidate genes of the brassinosteroid (BR) signaling pathway described above may play an important role in the defense against pathogens, which protects Mr5 against soil-borne pathogen attack and consequently increases the ARD tolerance.

## 4. Materials and Methods

### 4.1. Plant Material

Progenies of crosses between the apple rootstock ‘M9’ (mother) and the wild apple genotype Mr5 accession MAL0991 (father) as well as both parents were analyzed. The progeny were established in 2016 and 2019 and about 200 seeds of these crossings were sowed in vitro on MS medium [67]. After successful germination, seedlings were grown at 24 °C with 16 h of light provided by Philips MASTER TL-D 58 W/865 fluorescence tubes at a light intensity of 35–40 µmol m^−2^ s^−1^. When the first leaves developed, 1 mg leaf material of each seedling was collected and freeze-dried for DNA extraction. The progeny were further propagated in vitro in order to get 24 genetically identical replicates of each seedling for the phenotypic evaluation. Roots were induced after transferring five-week-old shoots to ½ MS medium supplemented with 2% sucrose and 4.92 µM IBA (no. M0222, Duchefa, Haarlem, The Netherlands). For acclimatization to greenhouse conditions, all plants were transferred to peat substrate (Steckmedium, Klasmann-Deilmann GmbH, Geeste, Germany) and cultivated under covers to ensure high humidity. During acclimatization, plants were adapted to greenhouse conditions by gradually reducing the air humidity. The plants were phenotyped in the greenhouse approximately four weeks post acclimatization.

### 4.2. DNA Extraction and SSR Genotyping

The dried leaf material was ground in a mixer mill apparatus (Retsch, Haan, Germany) and DNA isolation followed immediately after using the DNeasy Plant Mini Kit (Qiagen, Hilden, Germany) according to the manufacturer’s instructions. The DNA quantity was estimated by comparing a dilution series of 10, 20, 30, 40, and 50 ng of λ–DNA using the Quantity One*^®^* software, version 4.6 of the gel documentation system (Biorad, Feldkirchen, Germany). All samples were diluted to 10 ng/µL.

For SSR analysis, 111 SSR markers distributed over the 17 chromosomes of the *Malus × domestica* genome (Appendix A) were selected. Multiplex PCR was performed using the Type-It kit (Qiagen, Hilden, Germany) according to the manufacturer’s protocol, with up to six primer pairs per PCR in a total volume of 10 µL with the following conditions: 95 °C for 5 min, followed by 30 cycles of 95 °C for 1 min, 60 °C for 1 min 30 s, 72 °C for 30 s, and a final extension for 30 min at 60 °C. PCR fragments were analyzed on a 3500 Genetic Analyzer System (Thermofisher, Braunschweig, Germany), for which forward primers were labelled with fluorescent dyes.

### 4.3. Genotyping-by-Sequencing

A total of 140 progenies of the ‘M9’ × Mr5 cross and both parent genotypes were genotyped by tGBS^®^ genotyping-by-sequencing technology using an Illumina HiSeq X instrument (Data2Bio, Ames, IA, USA). Raw sequence data and reads were assigned to their corresponding samples before performing quality trimming to remove low-quality regions at the beginning and end of each read.

#### 4.3.1. Trimming and Alignment of Sequence Reads

Each individual sequence read was checked for low-quality sequences indicated by a PHRED quality score <15 (out of 40) [68], which corresponds to an estimated error rate of <3%. The remaining nucleotides were then scanned using overlapping windows of 10 bp, and sequences beyond the last window with average quality values of less than the specified threshold were truncated.

#### 4.3.2. SNP Discovery

Polymorphism at each potential SNP site was carefully examined, and putative homozygous and heterozygous SNPs were identified using different criteria for homozygous and heterozygous SNP calling. According to Data2Bio, an SNP call is homozygous if at least 5 reads supported the major common allele at that site and at least 90% of all aligned reads covering that site shared that same nucleotide at that site. An SNP call is heterozygous if at least two reads supported each of at least two different alleles and each of the two allele types separately comprised more than 20% of the reads aligning to that site. The sum of the reads supporting those two alleles should be at least equal to five and comprise at least 90% of all reads covering the site.

Subsequently, additional cut-offs were applied to improve the utility of the selected sets of SNPs. These filtering criteria were a minimum calling rate >50% (MCR50), allele number = 2, number of genotypes >2, minor allele frequency >10%, and heterozygosity rate range: 35–65%.

### 4.4. Construction of Genetic Maps

The genetic maps were constructed independently for each parent using the JoinMap v.5 program [69] with a logarithm of odds (LOD) of 5.0 or higher. All segregating SSR and SNP markers that showed polymorphism in at least one parent were used for mapping. Marker segregation ratios were calculated using the chi-square test. Markers with significantly distorted segregation (*p* < 0.05) were excluded from mapping. Linkage analysis was performed based on the regression mapping algorithm of the interval mapping approach [70]. Map distances were calculated in centiMorgans (cM) according to the Kosambi mapping function [71].

### 4.5. Phenotypic Evaluation and Marker–Phenotype Association Analyses

ARD was induced in the greenhouse by planting the seedlings of the progeny in ARD-infected soil. Due to the large number of plants to be tested, the phenotypic evaluation was divided into single tests that took place between 2017 and 2021. In each test, about 33–35 genotypes of the progeny were tested. Each test was repeated once, so that a total of 8 tests were carried out.

For the greenhouse test, ARD-affected soil from the experimental orchard of JKI Dresden-Pillnitz (51°00′01.6″ N 13°53′14.7″ E, Dresden, Germany) was collected as described in Reim et al. [31]. Half of the soil volume was γ-irradiated with a minimum dose of 10 kGy (recorded dosages: minimum 10.87 kGy, maximum 31.96 kGy, Steris, Köln, Germany) by which most fungi, bacteria, and invertebrates are killed [72]. Hereafter, the non-irradiated ARD-affected soil will be denoted as ARD soil and the γ-irradiated, disinfected ARD soil as γ-ARD soil.

Two to twelve replicates per genotype were planted into ARD soil or γ-ARD soil and cultivated for 8 weeks in a greenhouse, as described in Reim et al. [31]. For phenotyping, each individual plant was analyzed for shoot length increase. This was calculated by subtracting the shoot length at the end of the experiment (week 8) from the shoot length measured at the beginning of the experiment (week 0). For each seedling and treatment, the average shoot increase was calculated using the software program SASv.9. Subsequently, the average shoot increase (ASI) was calculated to assess the relative ARD reaction of each seedling of the progeny, as described in Reim et al. [31].

ASI values for only 106 progenies were employed for marker–phenotype association (Kruskal–Wallis) analysis because the rest of the plants had died during the experiment or showed no growth, and plants with less than three repetitions per genotype and variant were excluded from the study. Kruskal–Wallis analysis and QTL mapping were performed with MapQTL v.5 [73].

### 4.6. Identification of Potential Candidate Genes Linked to Markers

To identify potential candidate genes for ARD tolerance, linkage groups containing markers that were significant (*p* < 0.005) following Kruskal–Wallis analysis were examined in more detail. Therefore, the sequences of significant SNPs and SSR markers were aligned to the *Malus domestica* GDDH13 v.1.1 reference genome [33] using Galaxy (https://usegalaxy.org/, accessed on 1 March 2023) to identify the genes near the significant SNP/SSR. The genes in the closest proximity to the SNP/SSR markers were then selected as candidate genes.

## 5. Conclusions

The genetic mapping approach allowed for the identification of significant markers and regions in the *Malus* genome that are potentially associated with ARD tolerance/susceptibility. Blast alignment of the sequences of de novo SNP markers resulted in the identification of putative genes, whose in-depth analyses have revealed the possible pathways involved in this very complex trait. All candidate genes were associated with pathogen defense and encoded transcription factors or transcriptional repressors, functioned as receptors, or were involved in the regulation of various cellular processes. These genes were involved either in plant innate immunity or in a basal response to the pathogen due to a change in hormone synthesis. These findings will present opportunities for future studies, which will aim to elucidate the molecular defense mechanisms of ARD tolerance and susceptibility.

## Figures and Tables

**Figure 1 ijms-24-06307-f001:**
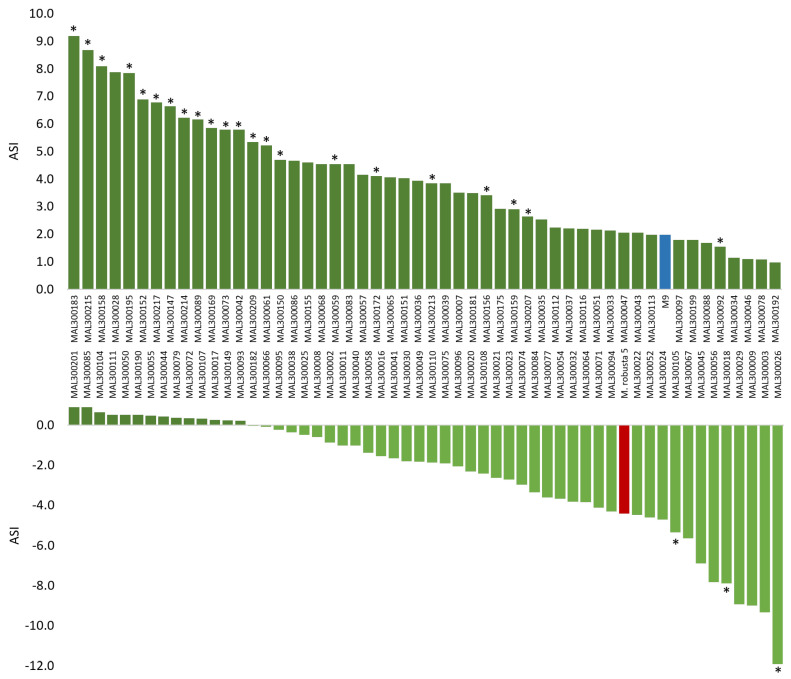
ARD susceptibility index (ASI) based on the shoot length differences calculated between plants cultivated either on ARD soil or on γ-ARD soil. Offspring with lower ASI values are less sensitive to ARD (light green bars). Offspring with an ASI value of zero or higher are more susceptible to ARD (dark green bars). Offspring with significant differences of the ASI (*p* < 0.05) were marked with *. The ARD-tolerant parent Mr5 showed an ASI value of −4.4 (red bar), whereas the ARD-susceptible parent ‘M9’ showed an ASI value of 2.0 (blue bar).

**Figure 2 ijms-24-06307-f002:**
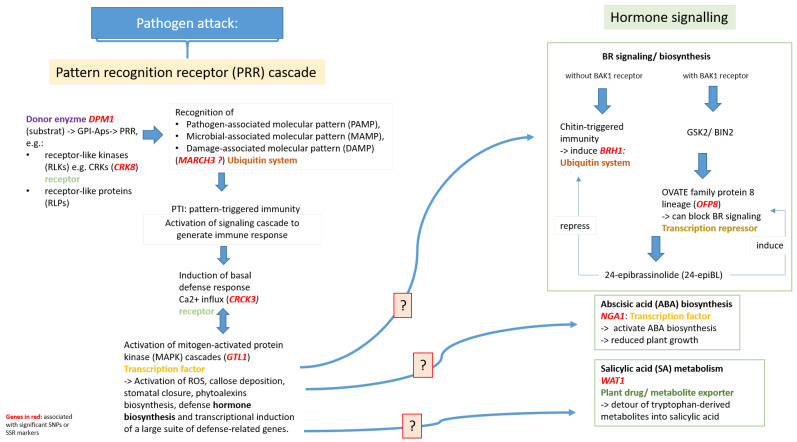
Candidate genes involved in the signal transduction pathway of plant innate immunity or subsequent hormone signaling in response to ARD. Candidate genes associated with significant SNP or SSR markers are highlighted in red. C*RK3*: *calmodulin-binding receptor-like cytoplasmic kinase 3*, *NGA1: B3 domain-containing protein Os03g0120900-like*, *MARCH3: E3 ubiquitin-protein ligase MARCH3*, *DPM1: dolichol-phosphate mannosyltransferase*, *WAT1: walls are thin1-related protein*, *GTL1: trihelix transcription factor*, *OFP8: OVATE family protein 8-line*, *CRK8: cysteine-rich RLK (RECEPTOR-like protein kinase) 8*, *BRH1: E3 ubiquitin-protein ligase RHA2B-like*.

**Table 1 ijms-24-06307-t001:** SSRs and SNPs mapped across the 17 linkage groups of ‘M9’ and Mr5.

	‘M9’			Mr5		
LG	Length (cM)	SNP Marker (*n*)	SSR Marker (n)	Length (cM)	SNP Marker (*n*)	SSR Marker (*n*)
LG1	64.83	65	3	59.94	94	4
LG2	82.03	109	6	52.10	176	5
LG3	70.36	132	5	56.29	246	5
LG4	52.95	126	1	66.69	114	4
LG5	80.87	131	7	86.15	83	7
LG6	73.19	50	3	89.61	266	6
LG7	78.81	100	3	63.53	104	4
LG8	56.81	122	0	54.95	129	0
LG9	65.38	189	3	60.56	165	3
LG10	84.35	139	2	72.30	178	2
LG11	94.15	94	2	54.04	267	3
LG12	63.51	88	2	55.21	199	2
LG13	60.51	107	3	62.92	322	4
LG14	65.12	127	0	52.93	186	1
LG15	106.86	115	5	76.33	180	4
LG16	76.87	116	4	75.18	149	5
LG17	77.32	124	6	58.96	131	5
sum	1253.92	1879	55 *	1097.70	2925	64 *

* SSR markers may have been integrated into both the maternal and paternal maps.

**Table 2 ijms-24-06307-t002:** Summary of the significant SNP and SSR markers detected by the Kruskal–Wallis test (KW) and their position on the linkage group for Mr5.

Marker	LG	Start	End	‘M9’	Mr5	Position ^1^	K* ^2^	Signif. ^3^
SNP_31883	6	26,471,091	26,470,957	G	G/T	52.66	8.01	****
SNP_32246	6	26,701,927	26,701,806	T	T/A	52.31	7.94	****
SNP_36646	6	26,470,806	26,470,950	A	A/G	52.78	8.89	****
SNP_36649	6	26,470,844	26,470,904	A	A/C	52.82	8.57	****
SNP_49388	6	26,982,732	26,982,590	T	T/C	52.41	8.08	****
SNP_54395	6	26,733,503	26,733,640	G	G/T	52.73	8.46	****
SNP_56130	6	26,648,239	26,648,380	C	C/G	52.13	8.14	****
SNP_56298	6	24,029,075	24,028,955	C	C/G	56.74	8.06	****
SNP_56970	6	26,705,196	26,705,318	A	A/T	52.63	8.27	****
SNP_57966	6	26,733,715	26,733,603	C	C/T	52.85	8.08	****
SNP_58944	6	24,726,042	24,726,172	A	A/T	55.97	8.98	****
SNP_58943	6	24,726,042	24,726,168	C	C/T	55.78	8.24	****
SNP_52341	2	1,549,119	1,548,999	G	G/A	8.31	8.62	****
Ch02c02a	2	3,716,747	3,716,507	178/192	0/184 ^###^	75.94 ^#^/0 ^##^	12.95	****

^1^ Position of the locus from the upper part of the linkage group in cM; ^#^ ‘M9’; ^##^ Mr5; ^###^allele conferring tolerance; ^2^ Kruskal–Wallis *K**; ^3^ Kruskal–Wallis Significance level at *p* = 0.005 (****).

**Table 3 ijms-24-06307-t003:** Putative candidate genes associated with the significant SNP markers and the SSR marker Ch02C02a and their respective functions.

Gene ID	LG	Start	End	Marker	Start	End	NCBI	Tair	Symbol	Function
MD02G1046400	2	3,703,811	3,705,009	Ch02c02a	3,716,747	3,716,507	XP_008386472	AT5G19650.1	OFP8	Transcription repressor OFP8-like [*Malus domestica*]
MD02G1022200	2	1,548,373	1,552,372	SNP_52341	1,549,119	1,548,999	XP_008385188	AT2G11520.1	CRCK3	Calmodulin-binding receptor-like cytoplasmic kinase 3 [*Malus domestica*]
MD06G1103200	6	24,022,662	24,023,312	SNP_56298	24,029,075	24,028,955	XP_008351649	AT5G41810.2	LOC103415066	Uncharacterized protein LOC103415066 [*Malus domestica*]
MD06G1107200	6	24,688,790	24,691,604	SNP_58943	24,726,042	24,726,168	CAN80145	AT4G23160.1	CRK8	Cysteine-rich RLK (RECEPTOR-like protein kinase) 8
	6			SNP_58944	24,726,042	24,726,172				
MD06G1123000	6	26,462,265	26,462,819	SNP_36646	26,470,806	26,470,950	XP_008358490	AT3G61460.1	BRH1	E3 ubiquitin-protein ligase RHA2B-like [*Malus domestica*]
	6			SNP_36649	26,470,844	26,470,904				
	6			SNP_31883	26,471,091	26,470,957				
MD06G1124400	6	26,645,912	26,650,494	SNP_56130	26,648,239	26,648,380	XP_008375464	AT2G46870.1	NGA1	B3 domain-containing protein Os03g0120900-like [*Malus domestica*]
MD06G1124900	6	26,701,497	26,703,758	SNP_32246	26,701,927	26,701,806	XP_008375454	AT2G45530.1	MARCH3	E3 ubiquitin-protein ligase MARCH3 isoform X2 [*Malus domestica*]
MD06G1125000	6	26,704,481	26,707,546	SNP_56970	26,705,196	26,705,318	XP_008375452	AT1G20575.1	DPM1	Dolichol-phosphate mannosyltransferase subunit 1 [*Malus domestica*]
MD06G1125400	6	26,731,654	26,733,668	SNP_54395	26,733,503	26,733,640	XP_008375451	AT3G30340.1	WAT1	WAT1-related protein At3g30340 isoform X2 [*Malus domestica*]
	6			SNP_57966	26,733,715	26,733,603				
MD06G1127800	6	26,980,060	26,984,800	SNP_49388	26,982,732	26,982,590	XP_008375438	AT1G76880.1	GTL1	Trihelix transcription factor GTL1 isoform X1 [*Malus domestica*]

Grey: Within the candidate gene sequence.

## Data Availability

Not applicable.

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
