# Peer review of "Deciphering the Mechanism of Tolerance to Apple Replant Disease Using a Genetic Mapping Approach in a Malling 9 × M. × robusta 5 Population Identifies SNP Markers Linked to Candidate Genes"

_ijms, 2023, doi:10.3390/ijms24076307_

Round 1

Reviewer 1 Report

In the submitted manuscript by Stefanie Reim, Ofere Francis Emeriewen, and colleagues entitled “Deciphering the mechanism of tolerance to apple replant disease using a genetic mapping approach in a Malling 9 × M. ×robusta 5 population identifies SNP markers linked to candidate genes”, the authors proposed some candidate genes that may offer tolerance in Apple Replant Disease (ARD). To achieve this goal a high-density genetic map using tunable genotyping-by-sequencing (tGBS) approach was applied. Overall, the manuscript is well-written, and the figures have a good presentation. The aim and scope of the International Journal of Molecular Sciences are in line with the current manuscript. However, there are two issues that should be carefully addressed.

Major and minor comments,

Lines 463-475: Please, define the range of replicates instead of ‘up to twelve replicates per genotype’.

Please provide as a supplementary table the mean, SD, and the number of replication (N) of the average shoot increase (ASI) of 106 progenies.

Line 153: Incorporate the bibliography ‘Error! Reference source not found.’

Author Response

Dear Reviewer,

We greatly appreciate your work on our manuscript. We thank you for your detailed and careful review and for the useful suggestions to improve the article. We have revised the article according to your comments to improve the evaluation and documentation of the results of our scientific analyses.

You may please find our responses to each comment in blue font.

Lines 463-475: Please, define the range of replicates instead of ‘up to twelve replicates per genotype’.

Thank you for your comment; we changed this sentence according to your suggestions.

Please provide as a supplementary table the mean, SD, and the number of replication (N) of the average shoot increase (ASI) of 106 progenies.

Thank you for your comment; this table was already provided (Table_S1_Output_Phenotyping). Maybe it was not uploaded?

Line 153: Incorporate the bibliography ‘Error! Reference source not found.’

Sorry for this. Sometimes this error message occurred instead of the reference to the table/figure when converting word to pdf. We revised this.

Reviewer 2 Report

The manuscript includes a lot of information, clearly raised issues, lack of knowledge and attempts to answer the goals set for themselves. Minor corrections required. You should also check in the reference whether all the sources are really needed, especially if we are talking about more than a decade old. 

Line 153 " Error! Reference source not found."

Author Response

Dear Reviewer,

We greatly appreciate your work on our manuscript. We thank you for your detailed and careful review and for the useful suggestions to improve the article. We have revised the article according to your comments to improve the evaluation and documentation of the results of our scientific analyses.

You may please find our responses to each comment in blue font.

The manuscript includes a lot of information, clearly raised issues, lack of knowledge and attempts to answer the goals set for themselves. Minor corrections required. You should also check in the reference whether all the sources are really needed, especially if we are talking about more than a decade old. 

We fully agree that some of the references are more than 10 years old and we deleted some of them. Most of the old references are related to studies on ARD, such as studies on resistant rootstocks or wild apple species. Since there are not so many recent studies and especially findings on this topic, we would like to keep these references.

Line 153 " Error! Reference source not found."

Sorry for this. Sometimes this error message occurred instead of the reference to the table/figure when converting word to pdf. We revised this. 
